# In Vitro and In Silico Antioxidant Activity of Novel Peptides Prepared from *Paeonia Ostii* ‘Feng Dan’ Hydrolysate

**DOI:** 10.3390/antiox8100433

**Published:** 2019-10-01

**Authors:** Min Wang, Cong Li, Haoyu Li, Zibo Wu, Bang Chen, Yibo Lei, Yehua Shen

**Affiliations:** 1Key Laboratory of Synthetic and Natural Functional Molecule Chemistry of Ministry of Education, College of Chemistry and Materials Science, National Demonstration Center for Experimental Chemistry Education, Northwest University, Xi’an 710127, Shaanxi, China; wm253340965@163.com (M.W.); yasylhy@163.com (H.L.); chinawu@protonmail.com (Z.W.); chenbang@nwu.edu.cn (B.C.); leiyb@nwu.edu.cn (Y.L.); 2College of Chemistry and Chemical Engineering, Yan’an University, Yan’an 716000, Shaanxi, China

**Keywords:** *Paeonia ostii* hydrolysate, peptide purification, antioxidant activity, DFT, structure-activity relationship, toxicity

## Abstract

Antioxidant peptides derived from natural products have superior performance and broader application prospects. In this study, five novel antioxidant peptides were prepared from *Paeonia ostii (P. ostii)* seed meal, moreover the bioactive and the relationship between structure and properties of antioxidant peptides were elucidated by quantum chemical calculations. The free radical-scavenging activities were used as indexes to purify and concentrate the antioxidant peptides through five proteases and separation techniques. FSAP (Phe-Ser-Ala-Pro), PVETVR (Pro-Val-Glu-Thr-Val-Arg), QEPLLR (Gln-Glu-Pro-Leu-Leu-Arg), EAAY (Glu-Ala-Ala-Tyr) and VLRPPLS (Val-Leu-Arg-Pro-Pro-Leu-Ser) were identified by nano liquid chromatography–tandem mass spectrometry (LC-MS/MS). In vitro antioxidant activity test, EAAY exhibited the highest 2, 2’-azino-bis (ABTS) and hydroxyl radical-scavenging activity of 98.5% ± 1.1% and 61.9% ± 1.3%, respectively (*p* < 0.01), at 0.5 mg/mL. In silico calculations were carried out using the density functional theory (DFT) with the B3LYP/6-31G* basis set. According to natural bond orbital (NBO) analysis, the bioactivity of free-radical scavenging of the peptides was presumed. Moreover, the antioxidant peptides demonstrated no obvious cytotoxicity to L929 fibroblast cells. Therefore, the peptides from *P. ostii* seed by-products might potentially have excellent uses in functional foods, nutraceuticals and pharmacological products.

## 1. Introduction

Oxidative stress is an imbalance between oxidation and antioxidation, causing potential damage to the body to a certain extent by the formation of excess free radicals, such as hydroxyl radical (OH^●^), superoxide anion (O_2_^−^^●^), and reactive oxygen species (ROS) [1]. The oxidative stress process can directly cause DNA double-strand cross-linking cleavage, intracellular protein and enzyme denaturation, and biofilm lipid peroxidation, leading to diseases such as aging, cancer, and Alzheimer’s [2,3,4]. With the attention paid to people’s health and the importance of dietary safety, the research on high-efficiency and safe natural antioxidants has been the focus of researchers at home and abroad. According to reported studies, the antioxidant peptide extracted from the natural products has free radical-scavenging activity, with less toxicity and fewer side effects on the human body than the chemically synthesized drug [5]. The preparation of novel antioxidant peptides and the structure–activity relationship have received increasing attention.

Based on density functional theory (DFT), quantum chemistry can perform effective theoretical calculations on the structure–activity of molecules through conformational optimization, the highest occupied molecular orbital (HOMO), the lowest unoccupied molecular orbital (LUMO) energy studies and natural bond orbital (NBO) analysis [6]. DFT was widely used in the fields of flavonoid compounds [7], metal elements, small organic molecules, drugs, and so on [8,9]. This method is rarely used in the research of the peptide field. Antioxidant polypeptides prepared from frog skin [10] were optimized for conformation using this method. Lin. et al. [11] evaluated the antioxidant activity of specific amino acids according to the BDE (bond dissociation energy), ionization potential (IP), proton affinity (PA) and so on. However, the use of NBO to speculate on the bioactivity of peptides from a natural plant has rarely been reported.

Peony, the Paeoniae family, is well known as a traditional Chinese ornamental, medicinal, and edible plant [12]. Because peony has important internal and external value, there is a tremendous number of peony species cultivated across the world [13]. As one of the peony varieties, *Paeonia ostii* ‘Feng Dan’ has been widely planted and considered as an economic plant. The effective ingredients extracted from different tissues of *Paeonia ostii* (*P. ostii*) have potentially high nutritional value, and biological activity [14]. Paeonol extracted from its root bark has effective medicinal properties such as anti-inflammatory, antibacterial, hemostasis and antihypertensive properties [15]. What is more, several studies have shown *P. ostii* seed oil is considered to be functional edible oil that is rich in α-linolenic acid (>40%) and polyunsaturated fatty acids (>90%) [12]. With the increasing market demand of *P. ostii* seed oil, large amounts of defatted peony seed meal are produced as waste and can be used as the material of antioxidant peptides.

In this paper, *P. ostii* seed meal protein (*Po*SMP) was hydrolysed using different peptidases. The antioxidant activity of *P. ostii* seed peptides, including 1, 1-diphenyl-2-picrylhydrazyl (DPPH), hydroxyl, 2, 2’-azino-bis-(3-ethylbenzothiazoline-6-sulfonic acid) (ABTS), and superoxide anion radical-scavenging activity were evaluated. Five novel antioxidant peptides were separated and identified. The quantum chemistry was used to evaluate bioactive and activity magnitudes of the novel peptides based on charge distribution and frontier molecular orbital.

## 2. Materials and Methods

### 2.1. Reagents

*P. ostii* ‘Feng Dan’ seeds were collected from Bozhou, Anhui, China. DPPH, ABTS were purchased from Sigma-Aldrich (Saint Louis, MO, USA). Marker (PR1910: 11–180 kDa) was purchased from Solarbio^®^ life science company (Beijing, China). Ascorbic acid and glutathione (GSH) were purchased from Aldrich Chemical Co., Ltd. (Shanghai, China). Acetonitrile, methanol, formic acid (FA), trifluoroacetic acid (TFA) and dimethyl sulfoxide are chromatographic grade reagents. L929 fibroblast cells were provided by Xi’an Jiaotong University. All other chemicals and reagents were of analytical grade. Water was obtained from a Milli-Q system from Millipore (Bedford, MA, USA).

### 2.2. Extraction of Bioactive Peptides from P. Ostii Seed Meal Protein (PoSMP)

#### 2.2.1. Preparation of *Po*SMP

The *Po*SMP was prepared using a previously described method [16]. The *P. ostii* seed meals were obtained by physical cold pressing technology and further degreased with n-hexane a ratio of 1:5 (*w*/*v*). The defatted *P. ostii* seed powder was dissolved in deionized water at a ratio of 1:10 (*w*/*v*), and the pH was adjusted to 10.0 using 1.0 M sodium hydroxide. After continuous stirring for 4 h, the dispersion was centrifuged at 7000 rpm for 15 min. Then, the pH of supernatant was adjusted to 4.5 with 1.0 M hydrochloric acid and the precipitate was adjusted to neutral pH, The *Po*SMP was lyophilized and stored at −80 °C for further hydrolysis. The protein content of obtained *Po*SMP was 81.6% ± 0.48% measured by the Lowry method.

#### 2.2.2. Preparation of Protein Hydrolysates from *Po*SMP

*Po*SMP was dispersed in distilled water at a concentration of 40 g/L in 85 °C for 20 min. The optimized conditions for five food-grade enzymes are displayed in Table 1. When the pH and temperature of the solution were stabilized, the enzyme was added at a protein-to-enzyme ratio of 25:1. Each enzyme solution was adjusted to pH ± 0.2 with 0.5 M NaOH or HCl during the hydrolysis. After the solution was incubated at boiling water for 10 min to terminate the enzymatic reaction, the pH was adjusted to neutral, and the mixture was centrifuged at 10,000 rpm for 10 min. The supernatant, *P. ostii* seed meal protein hydrolysates (*Po*SMPHs), was collected for further study on the degree of hydrolysis and antioxidant activity. The pH-stat method was used for the final determination of the degree of hydrolysis (DH) [17].

### 2.3. Sodium Dodecyl Sulphate Polyacrylamide Gel Electrophoresis (SDS-PAGE)

Sodium dodecyl sulphate polyacrylamide gel electrophoresis (SDS-PAGE) analysis of the *Po*SMP and protein hydrolysates were conducted using a discontinuous buffer system on a 20% separating gel and 5% stacking gel [18]. The samples at a concentration of 2 mg/mL were treated with 2 × SDS buffer solutions and boiled at 100 °C for 5 min. Next, the treated sample solutions were loaded per well and run at 80 V for 30 min on the stacking gel and 120 V on the separating gel. After dyeing in Coomassie brilliant blue G-250 staining solution and decolorizing the gel with a decolouring in distilled water: ethanol: glacial acetic acid, 9:3:1 (*v*/*v*/*v*), solution, the picture was captured using gel imaging system, OmegaLumC (RING MAN Biological Technology Co., Ltd., Shanghai, China).

### 2.4. Purification and Identification of the Bioactive Peptides Extracted from PoSMP

#### 2.4.1. Ultrafiltration

Alcalase hydrolysates (AHs) were separated by the Amicon^®^ stirred cells ultrafiltration system from Millipore (Ufsc40001, EMD Millipore Corporation, Billerica, USA), using a membrane (PLAC0710, EMD Millipore Corporation, Billerica, USA) having a molecular weight (MW) cutoff of 10, 5 and 1 kDa. The solution was divided into four sections: F1 (MW > 10 kDa), F2 (10 kDa > MW > 5 kDa), F3 (5 kDa > MW > 1 kDa) and F4 (MW < 1 kDa). Each fraction was freeze-dried and stored at −80 °C for further study.

#### 2.4.2. Ion-Exchange Chromatography

The component of MW <1 kDa with the highest antioxidant activity was purified using ion-exchange chromatography. A HiTripTM Q HP anion exchange column (GE, Boston, USA) was selected to isolate and purify fraction F4 at a concentration of 30 mg/mL using the AKTA Pure System CU-950 (GE, Boston, USA). The anion exchange column was equilibrated with 20 mM Tris-HCl buffer (pH 6.8). The sample was eluted in a linear gradient of 1 M NaCl (0–100%) in a total volume of 150 mL at a flow rate of 2.0 mL/min and the elution was monitored at 280 nm. All fractions were collected and lyophilized for further study.

#### 2.4.3. Reversed-Phase High-Performance Liquid Chromatography (RP-HPLC)

The fraction after HiTripTM Q HP anion exchange column with the highest antioxidant activity was dissolved in eluent A (ultrapure water containing 0.1% TFA) and further separated using reversed-phase high-performance liquid chromatography (RP-HPLC) on a ZORBAX C18 column (4.6 mm × 250 mm, 5 μm, Agilent Technologies Inc., USA). The sample solution was eluted in linear gradient that 5–60% B (0.1% TFA in acetonitrile) with a flow rate of 1 mL/min, and ultraviolet (UV) wavelength detection was performed at 220 nm. Each peptide fraction was collected and freeze-dried, and the antioxidant activity was determined.

#### 2.4.4. Nano Liquid Chromatography–Tandem Mass Spectrometry (LC-MS/MS) Analysis

The peptide fraction with the highest antioxidant activity after separation on the semi-preparative C18 column was re-suspended in buffer A (0.1% FA in deionized water). The online nano-RPLC (nano-reversed-phase liquid chromatography) was employed on the Easy-nLC 1200 System (Thermo Scientific, Rockford, USA). Peptides were separated on an analytical column (Acclaim PepMapTM RSLC, C18 50 μm × 150 mm 2 μm NanoViper, Thermofisher, Rockford, USA) at 120 min gradients: from 2% to 20% buffer B (0.1% FA in acetonitrile) in 90 min, from 20% to 30% B in 15 min, from 30% to 90% buffer B in 3 min and the remaining at 90% for 10 min. Data acquisition was performed with a Q Exactive Plus System (Thermo Scientific, Rockford, USA) fitted with a nanospray. The Q Exactive Plus instrument was operated using a data-dependent top-20 experiment with 70 K resolution for the full MS scans, 17.5 K resolution for high-energy collisional dissociation (HCD) tandem mass spectrometry (MS/MS) scans and a dynamic exclusion of 30 s and automatic gain control target of 105. Full MS scans were acquired in the Orbitrap mass analyser (ThermoFisher, San Jose, CA) over the range m/z 350–1500 with a mass resolution of 70,000 and an isolation window of 1.6 m/z. The twenty most intense peaks with a charge state ≥2 were fragmented in the HCD collision cell with normalized collision energy of 28%. Raw data were retrieved using Xcalibur (version 2.2, Thermo Fisher Scientific, Rockford, USA).

#### 2.4.5. Identification of Peptides

The raw MS/MS data was subjected to de novo sequencing using PEAKS software (version X, Bioinformatics Solutions Inc., Waterloo, Canada). A database search was performed on the results, which contained 5076 protein entries of *Paeonia* (downloaded from NCBI (National Center for Biotechnology Information) on Nov, 2018). The peptides were filtered based on the de novo only portion of the average local confidence (ALC > 90%) and b, y ion fragments that never matched the database. Oxidation (M, + 15.99) was set as a variable modification, and other search parameters used were as follows: precursor mass search type, monoisotopic; max variable PTM (post-translational modification) per peptide, 3; parent mass error tolerance, 5.0 ppm; fragment ion mass tolerance, 0.02 Da. In addition, contamination analysis of the raw data showed that the sample was free of contamination.

### 2.5. Synthesis of Antioxidant Peptides

According to the results of the nano liquid chromatography–tandem mass spectrometry (LC-MS/MS) analysis and identification, five novel peptides were obtained. Then five peptides were synthesized by the Peptides Co., Ltd. (Hefei, Anhui, China) using the solid phase synthesis method. The purity of the five synthesized peptides was higher than 98% by RP-HPLC analysis, and molecular mass was measured by LC-MS.

### 2.6. Antioxidant Capacity Assays

#### 2.6.1. DPPH (1, 1-Diphenyl-2-Picrylhydrazyl) Radical-Scavenging Assay

The radical-scavenging assay of DPPH was evaluated by a previously described method [19] with minor modification. In this assay, DPPH radical scavenging was evaluated based on the principle of the sample reducing the content of free radicals. For that purpose, 10 μL of the sample with 10 μL of 0.1 mM DPPH solution was dissolved in 95% ethanol, and the mixture was allowed to react in the dark for 30 min at room temperature. The solutions were analysed using a spectrophotometer at an absorbance of 517 nm. The DPPH radical-scavenging rate (1) was calculated as follows:DPPH.sc (%) = [A_0_ − (A − B)] / A_0_ × 100%(1)where A is the absorbance of sample; A_0_ is the absorbance of distilled water in place of the sample, which was measured under the same conditions; and 95% ethanol solution instead of DPPH was used to determine the absorbance value B. The same concentration of ascorbic acid was used as a positive control.

#### 2.6.2. Hydroxyl Radical-Scavenging Activity Assay

The assay to determine hydroxyl radical-scavenging activity was conducted as outlined previously [20] with slight modification. A sample of 100 μL was prepared mixed with 30 μL of H_2_O_2_ standard solution (6 mM) and 30 μL of Fe (II) solution (6 mM), then hydroxyl radicals were generated by the Fenton reaction. A coloured material was formed by adding 30 μL of salicylic acid-ethanol solution (6 mM) with hydroxyl radical, and 210 μL of distilled water was added to the reaction tube. Subsequently, the mixture was vortexed using a vortex mixer and incubated at 37 °C for 30 min. After the reaction, the solution was analysed using a spectrophotometer at an absorbance of 510 nm. The hydroxyl radical-scavenging activity (2) was calculated as follows:OH·sc (%) = [1 − (A − B) / A_0_] × 100%(2)where A is the sample absorbance, A_0_ is the blank control absorbance with distilled water as a reference, and B is the reagent without salicylic acid. The same concentration of ascorbic acid was used for the treatment and as a positive control.

#### 2.6.3. ABTS (2, 2’-Azino-bis) Radical-Scavenging Assay

The ABTS radical-scavenging assay was measured in accordance with a method described previously [21] with slight modification. Equal volumes of 7.4 mM ABTS solution and 2.6 mM potassium persulfate solution were mixed, reacted in the dark for 14–16 h at room temperature to form ABTS radical stock solution, and diluted with 10 mM phosphate-buffered saline (PBS, pH 7.4) so that the absorbance at 734 nm was 0.70 ± 0.02. Next, 20 μL of the sample was mixed with 480 μL of the ABTS radical solution and allowed to react in the dark at room temperature for 10 min. The absorbance was measured at 734 nm. The ABTS radical-scavenging rate was determined according to the following formula (3):ABTS^+^·sc (%) = (A_0_ − A) / A_0_ × 100%(3)where A is the absorbance of the sample, and A_0_ is the absorbance of distilled water instead of the peptide solution. GSH at a concentration 0.1 mg/mL was used as a positive control.

#### 2.6.4. Superoxide Anion Radical-Scavenging Activity

The radical-scavenging assay of superoxide anion radical (O_2_^−^^●^) was tested using a procedure described previously [22]. Two millilitres of Tris-HCl buffer solution (0.1 M, pH 8.0) was mixed with two millilitres of the sample solution at 2.0 mg/mL. The solution was incubated at 25 °C for 20 min. Subsequently, one millilitre of 3.0 mM pyrogallol solution in 1.0 mM HCl was preheated at 25 °C and added. The absorbance of the mixed solution was measured at 320 nm and recorded every 30 s for 4 min. Next, considering time as the abscissa and absorbance as the ordinate to draw a straight line, the pyrogallol self-oxidation inhibition rate (or superoxide anion radical-scavenging rate) formula (4) was used as follows:O_2_^−^^●^sc (%) = (K_0_ − Ks) / Ks × 100%(4)where the slope Ks is the inhibition effect of the sample on the auto-oxidation of pyrogallol, K_0_ is the distilled water instead of the sample to indicate the slope of auto-oxidation rate of pyrogallol. The same treatment of GSH at a concentration of 2 mg/mL was used as a positive control.

### 2.7. Quantum Chemical Calculations

All quantum chemical calculations in this study were performed using the Gaussian 09 system program [23]. We have employed the DFT method at B3LYP/6-31G* level to optimize, and optimized to virtual frequency without negative values. The atomic net charge distribution was obtained by the NBO 3.1 program [24] operation.

### 2.8. In Vitro Cytotoxicity Assay on L929 Fibroblast Cells

L929 fibroblast cells were chosen for the cytotoxicity assay, which were cultured in Dulbecco’s modified Eagle medium high glucose containing 10% fetal bovine serum (FBS), 1% penicillin/ streptomycin solution. L929 fibroblast cells were cultured in sterile 96-well plates at a volume of 200 μL per well at approximately 1.0 × 10^5^ cfu/mL. Next, the supernatant was removed, and cytotoxicity studies were performed with different concentrations (0, 0.2, 0.4, 0.6, and 0.8 mg/mL) of AH prepared in the medium and the incubation was conducted for 48 h. The cytotoxicity of the samples was determined at 490 nm using a microplate reader (Thermo scientific, VARIOSKAN FLSAH, California, USA) according to the thiazolyl blue tetrazolium bromide (MTT) method [25]. All experiment groups were in quintuplicate, *n* = 5.

### 2.9. Statistical Analysis

All the measurements were conducted in triplicate. All the results were presented as the mean ± standard deviation from the triplicates, *n* = 3. Data were processed by analysis of variance and mean separations were performed through the Duncan’s multiple range test, with reference to 0.05 probability level, using the SPSS 19 software (version 19, SPSS Inc., Chicago, IL, USA).

## 3. Results and Discussion

### 3.1. Hydrolysis of PoSMP with Peptidases and Antioxidant Activity

The extraction yield of *Po*SMP extracted from defatted *P. ostii* seed meal reached 46.7% ± 0.3% using the reported method [26]. To effectively release antioxidant peptides, a method using peptidase hydrolysis was adopted and five food-grade enzymes were used by comparison [27]. The DH of PoSMP of five enzymatic hydrolysates indicated a rapid increase in the DH during the first 30 min, and the DH showed a tendency to increase slowly after 4 h, and then stabilized. This trend may be due to the gradual decrease in the activity of enzyme during the hydrolysis of *Po*SMP. It is worth noting that the hydrolysis curve of neutrase hydrolysates is different from the other four (See Figure 1 (A)). After 6 h of hydrolysis, AHs showed the highest DH at 14.7% ± 0.2% (*p* < 0.05), followed by trypsin (12.8% ± 0.1%), papain (10.1% ± 0.3%), neutrase (9.6% ± 0.5%), and pepsin (7.9% ± 0.1%). The different DH of the hydrolysates showed that the enzyme had different cleavage sites on the *Po*SMP, which resulted in different peptide structures and functions [28].

The antioxidant activity is closely related to the size, charge and amino-acid composition of the peptides produced by hydrolysis [20]. The antioxidant activity of different enzyme hydrolysates is shown in Table 1. Pepsin hydrolysates have superior DPPH radical-scavenging activity with a clearance rate of 80.2% ± 1.8%. AHs possess the highest ABTS radical-scavenging activity of 45.5% ± 1.5% (*p* < 0.05) and hydroxyl radical-scavenging activity of 56.1% ± 1.8%, which were significantly superior to other peptidase hydrolysates at the concentration of 1 mg/mL. Moreover, AHs exhibited stronger superoxide anion radical-scavenging activity, which may be caused by the specificity of the enzymes or the inherently folded structure of *Po*SMP [29]. Overall, alcalase was more effective in hydrolysing *Po*SMP.

### 3.2. SDS-PAGE of the P. ostii Seed Meal Protein Hydrolysates (PoSMPHs)

The specific antioxidant activity of different enzymatic hydrolysis was intrinsically closely related to the molecular weight distribution of the peptide. The molecular weight distribution of *Po*SMP and five *Po*SMPHs were monitored by SDS-PAGE. Figure 1B demonstrates that the 23, 33 and 34 kDa proteins are the main components of *Po*SMP, followed by 38 and 49 kDa proteins. The 34 kDa band with the highest protein content was conveniently hydrolysed by alcalase and pepsin, which suggested that the band may have a large number of amino acids at the two enzyme cleavage sites or related to the hydrolysis condition [30]. In contrast, proteins at 23 and 38 kDa showed strong resistance to most of the selected enzymes, and only pepsin could hydrolyse under acidic conditions, which may be related to the spatial structure and special functions of the proteins [31]. However, the protein band of the pepsin hydrolysates (PeHs) was contradictory to the low degree of hydrolysis, which deserves further research and in-depth discussion.

### 3.3. Purification and Identification of the Antioxidant Peptides

#### 3.3.1. Purification of Alcalase Hydrolysates (AHs) using Ultrafiltration

To enrich peptide fractions with outstanding antioxidant activity, the alcalase hydrolysate was primitively separated by ultrafiltration [32]. As shown in Table 2, fraction F4 had the highest yield among the four fractions showed the most potent antioxidant activities, with an ABTS radical-scavenging activity of 69.7% ± 1.0% (*p* < 0.01), compared with fractions F1, F2, and F3, whose values were 47.4% ± 1.0%, 36.8% ± 1.2%, and 41.3% ± 1.7%, respectively. Next, the DPPH and super oxygen radical-scavenging activity were evaluated. From the results, fraction F2 exhibited the strongest super oxygen scavenging activity of 37.8% ± 1.3%, which was slightly higher than that of fraction F4, 30.8% ± 1.4%. As reported previously [33], the large-sized peptide fractions of barley glutelin hydrolysates exhibited higher DPPH radical-scavenging activity than the small-sized peptide fractions. Furthermore, the strongest hydroxyl radical-scavenging capacity was 68.8% ± 1.2% (*p* < 0.01), for fraction F4, relative to the other components. These results revealed that MW < 1 kDa peptides were advantageous in eliminating ABTS and hydroxyl radicals. From the above results, we concluded that fraction F4 was the major fragment with potential for antioxidant activity.

#### 3.3.2. Purification of F4 Using Anion Exchange Chromatography

The F4 fraction was segmented via a HiTripTM Q HP column, and a total of seven peaks were eluted, as shown in Figure 2A. Among them, fraction P3 exhibited stronger radical-scavenging activities (as shown in Figure 2B), with an ABTS scavenging activity of 82.3% ± 1.2% and a hydroxyl scavenging activity of 70.4% ± 1.7%. This fragment exhibited a slightly better radical-scavenging activity than the peptides of cherry seed that showed a hydroxyl radical-scavenging activity of 58.2% ± 0.89% and ABTS radical-scavenging activity of 55.58% ± 0.91% [26]. As the sodium chloride concentration continued to rise, the peptide components (P4–P7) were eluted gradually through the column, and the radical-scavenging rate was reduced sequentially at the same concentration. The results showed that the peptide fragment carrying an anionic group in the 20 mM Tris-HCl buffer solution (pH 6.8) had stronger binding ability to the free radical, which was related to the nature of the amino acid in the peptide sequence. Therefore, the fraction P3 was selected and collected for further isolation and purification.

#### 3.3.3. RP-HPLC of Fraction P3

As shown in Figure 2C, after separation by reversed-phase chromatography, the P3 component peptides were divided into nine intense peaks, namely, S1 to S9. The different fractions were collected separately and used to evaluate the antioxidant activities, as shown in Figure 2D. Fraction S2 possessed the highest ABTS scavenging activity of 93.5 ± 1.4% (*p* < 0.01) and hydroxyl scavenging activity of 72.5% ± 1.6% (*p* < 0.01). The ABTS radical-scavenging activity of the S2 fraction was slightly higher than the antioxidant activity of the peptide extracted from the hazelnut protein, such that the C2 fraction of ABTS scavenging activity was 92.9% ± 1.0% [19]. In contrast, as the polarity of the mobile phase gradually decreased, the peptide sequences with non-polar groups exited the column and exhibited a weaker free radical-scavenging capacity.

### 3.4. Identification and Characterization of Antioxidant Peptides

After the fraction S2 purification by RP-HPLC was collected, and the amino sequence was further identified by Nano LC-MS/MS. The PEAKS software identified five novel antioxidative peptides by de novo technology from *P. ostii* seed meals: FSAP (Phe-Ser-Ala-Pro, 421.2082), PVETVR (Pro-Val-Glu-Thr-Val-Arg, 699.3915), QEPLLR (Gln-Glu-Pro-Leu-Leu-Arg, 754.4337), EAAY (Glu-Ala-Ala-Tyr, 452.1907) and VLRPPLS (Val-Leu-Arg-Pro-Pro-Leu-Ser, 780.4857). The antioxidant activity is closely related to hydrophobic amino acids such as Pro, Ala, and Glu [34]. The five peptides contained aromatic amino acids such as Pro and Tyr, which can provide protons to scavenge free radicals. Furthermore, the simultaneous appearance of the same hydrophobic amino acids such as Ala-Ala, Leu-Leu, and Pro-Pro were pivotal indicators of the powerful free radical-scavenging activity of the peptides [21].

The five peptides were obtained by solid-phase synthesis and measured using the ABTS and hydroxyl radical-scavenging capacity assays. As we can see from Figure 3, the EAAY peptide sequence showed hydroxyl radical scavenging with 61.9% ± 1.3% at a concentration of 0.5 mg/mL. When the concentration was 1 mg/mL, the EAAY hydroxyl radical-scavenging activity reached 75.7% ± 1.3%. The hydroxyl radical-scavenging ability of the FSAP peptide sequence (45.0% ± 1.8%) and the VLRPDLS sequence (44.4% ± 1.6%) were similar at 1 mg/mL. The novel peptides from *Po*SMPHs had lower hydroxyl radical-scavenging activity than the antioxidant peptide sequence LAYLQYTDFETR (47.42% at 0.1 mg/mL) obtained from pecan meal hydrolysate [35]. At a 0.5 mg/mL concentration, the EAAY peptide sequence exhibited a similar ability as GSH to clear ABTS free radicals, which are 98.5% ± 1.1% and 99.2% ± 1.6%, respectively (*p* < 0.01). The ABTS radical-scavenging ability of the EAAY sequence was stronger than the ADGF (Ala–Asp–Gly–Phe) fragment (0.5 mg/mL), which was composed of the same amino-acid number from hazelnut protein hydrolysates [19]. Compared to the other four peptide sequences, EAAY showed the highest ABTS and hydroxyl radical-scavenging capacities. The hydrophobic amino acid Ala and the aromatic amino acid Tyr presented at the C-terminus may be the structural basis for EAAY to exhibit strong ABTS radical-scavenging capacity [36,37].

### 3.5. Computational Methodology

DFT theory can be used to explain the relationship between free radical scavenging and structure–activity of peptide molecules [11]. According to the theoretical level of DFT / B3LYP / 6-31G*, the chemical reaction sites and activity magnitudes of the five peptides were predicted and calculated by quantum chemical calculations. The HOMO, LUMO, and energy gap (∆E) of the five peptides are shown in Figure 4. The HOMO energy value characterizes the ability of electron-donating and the LUMO characterizes the ability of electron-accepting [6,8,9]. In addition, HOMO and LUMO energy is directly related to IP ≈ −EHOMO (negative value of HOMO energy), EA ≈ −ELUMO (negative value of LUMO energy), and chemical potential μ (≈ − (IP+EA)/2) respectively [38]. The energy gap of HOMO and LUMO reflects the biological activity of the molecule and small ∆E generally has a flexible chemical reactivity [39]. Therefore, the distribution sites of HOMO and LUMO have an extremely close relationship with the free radical-scavenging activity of the peptides. As shown in Figure 4, the HOMO is concentrated on the guanidyl in Arg of PVETVR and QEPLLR; the phenolic hydroxyl structure in Tyr of the EAAY; imino structure in Pro of VLRPPLS. In addition, LUMO is mainly distributed on the structure related to the carboxyl group.

The NBO is an effective tool for the analysis of natural charge distribution. It is well known that the difference in charge on a chemical bond determines the activity of the site [40]. Figure 5 demonstrated the structural diagram after semi-empirical and DFT optimization, with serial numbers on each atom. Combining optimized peptide conformational pictures, Table 3. demonstrated that the largest difference in the charge between the atoms is mainly concentrated in N1-H42 (1.27412) in amino group and O22-H23 (1.24234) in hydroxyl of FSAP; N53-H54 (1.28215) and N58-H59 (1.29907) in guanidyl of PVETVR and QEPLLR; N1-H47 (1.31684) in amino group and O36-H37 (1.17416) in phenolic hydroxyl of EAAY; O61-H62 (1.25088) in hydroxyl of VLRPPLS. Most noteworthy, the N1 in EAAY glutamic acid has the largest charge value (0.91883), which is higher than the maximum net negative charge value of the other four peptide molecules. To sum up, according to NBO analysis of in silico calculations, the phenolic hydroxyl structure of Tyr (EAAY), the hydroxyl structure of Ser (FSAP), and the guanidyl structure of Arg (PVETVR, QEPLLR) conveniently achieved the purpose of scavenging free radicals. In the preparation and isolation of antioxidant peptides from food-borne materials, the peptide sequences rich in Tyr, Ser and Arg (especially Tyr) need to be given more attention, as they provide a potential contribution for antioxidant activity.

### 3.6. In Vitro Cytotoxicity Assay on L929 Fibroblast Cells

Normal cells, L929 fibroblast cells, were used to detect the toxicity of the alcalase hydrolysates. Toxicity assay were performed on AHs at concentrations of 0, 0.2, 0.4, 0.6, and 0.8 mg/mL and the cell viability without the added sample was 100.00% ± 1.6%. At the same time, the cell viability reached a peak of 113.6% ± 2.8% when the concentration reached 200 μg/mL. As the concentration increased to 400 μg/mL, 600 μg/mL and 800 μg/mL, the cell viability decreased steadily to 109.8% ± 3.6%, 108.1% ± 2.1% and 102.2% ± 2.4%, respectively. But, these data were higher than the blanks without sample. The results indicate that the AHs did not exhibit significant cytotoxicity against L929 fibroblast cells.

## 4. Conclusions

In this experiment, antioxidant peptides were hydrolyzed from the *P. ostii* seed meal protein using five enzymes, and the degree of hydrolysis by alcalase was significantly better than other enzymes. After purification of alcalase hydrolysate, five novel antioxidant peptides FSAP, PVETVR, QEPLLR, EAAY and VLRPPLS were identified. Among them, the EAAY sequence demonstrated the significantly strongest ABTS and hydroxyl radical-scavenging ability. In addition, in silico calculations were carried out using the DFT with the B3LYP/6-31G* basis set. The amino group, sulfhydryl group and phenolic hydroxyl group (Tyr in EAAY) are momentous for scavenging free-radical reactions. Furthermore, L929 fibroblast cells experiments showed that the antioxidant peptides prepared from *P. ostii* seed meal had no significant cytotoxicity. Accordingly, the antioxidant components isolated from the *P. ostii* byproduct not only enhance the value of plant seed processing, but also have potential value in the field of functional foods and pharmacological fields.

## Figures and Tables

**Figure 1 antioxidants-08-00433-f001:**
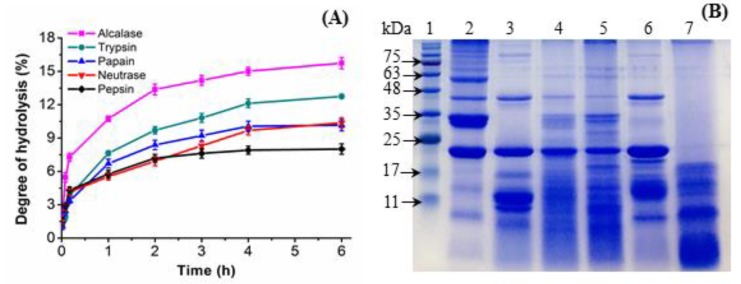
(**A**) Degree of hydrolysis of *P.*
*ostii* seed meal protein (*Po*SMP) by five different proteases. (**B**) Sodium dodecyl sulphate polyacrylamide gel electrophoresis (SDS-PAGE) of *Po*SMP and five peptidase hydrolysates. Lane 1: Marker (protein standard); lane 2: *Po*SMP; lane 3: AHs (alcalase hydrolysates); lane 4: THs (trypsin hydrolysates); lane 5: PaHs (papin hydrolysates); lane 6: NHs (neutrase hydrolysates); lane 7: PeHs (pepsin hydrolysates).

**Figure 2 antioxidants-08-00433-f002:**
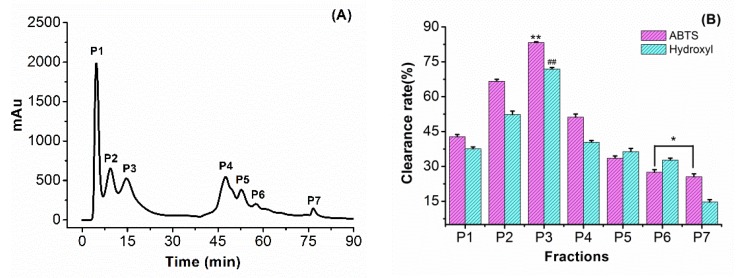
Separation and purification of antioxidant peptides and free radical-scavenging activity: (**A**) separation of peptide fractions (F4) on an anion exchange HiTrapTM Q HP column. (**B**) ABTS (2, 2’-Azino-bis) and hydroxyl radical-scavenging capability of different components (P1–P7) on a HiTrapTM Q HP column. (**C**) Separation of fraction P3 on C18 column. (**D**) ABTS and hydroxyl radical-scavenging capability of different fractions S1–S9. Results are shown as means ± standard deviation, *n* = 3; Duncan’s multiple range test was used for mean separation; *** p (##p)* < 0.01 represents the difference from other data in the group, ** p (#p)* < 0.05 represents the difference between the two sets of data.

**Figure 3 antioxidants-08-00433-f003:**
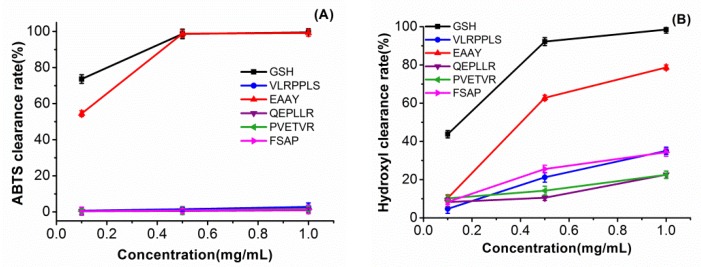
ABTS scavenging capacity of identified peptide (**A**) and hydroxyl scavenging capacity of identified peptide. (**B**) All experiments were in triplicate, *n* = 3.

**Figure 4 antioxidants-08-00433-f004:**
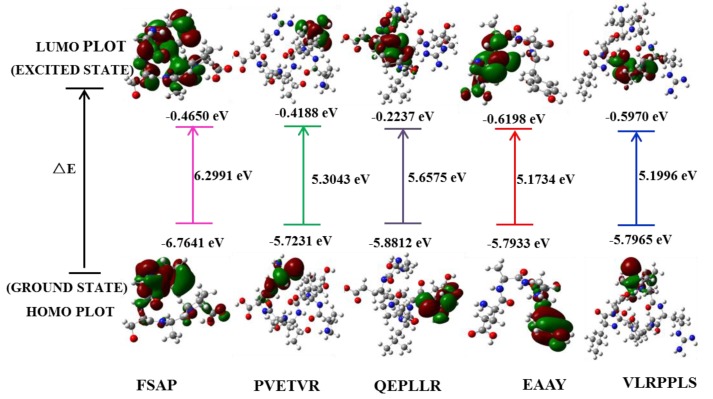
The frontier molecular orbital of five novel peptide molecules: the data on the line and below the line represent the lowest unoccupied molecular orbital (LUMO) and the highest occupied molecular orbital (HOMO) value, respectively. The data between the lines represents the LUMO and HOMO energy gap values expressed as ΔE, and the unit is expressed by eV.

**Figure 5 antioxidants-08-00433-f005:**
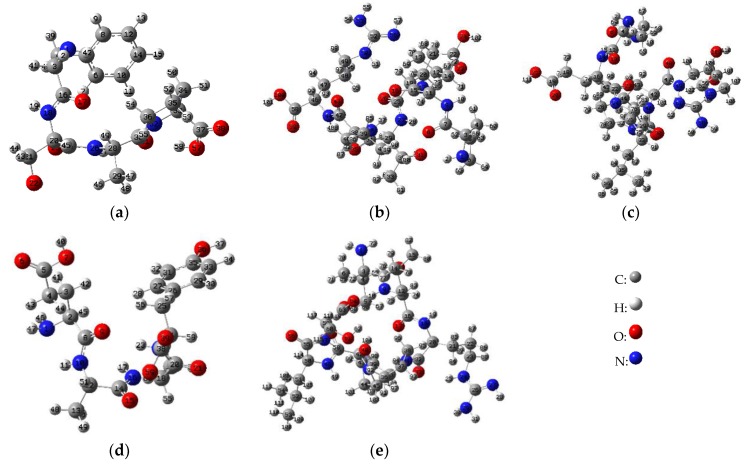
Optimized five peptides molecular structure with numbering of atoms and the atomic number of the molecule is derived from Gaussian 09 system. (**a**) to (**e**) are FSAP (Phe-Ser-Ala-Pro, 421.2082), PVETVR (Pro-Val-Glu-Thr-Val-Arg, 699.3915), QEPLLR (Gln-Glu-Pro-Leu-Leu-Arg, 754.4337), EAAY (Glu-Ala-Ala-Tyr, 452.1907) and VLRPPLS (Val-Leu-Arg-Pro-Pro-Leu-Ser, 780.4857), respectively.

**Table 1 antioxidants-08-00433-t001:** The optimal conditions for the hydrolysis of *Po*SMP by different enzymes and the antioxidant activity of the different enzymatic hydrolysates.

Enzyme (activity)	Proteolytic activity of enzyme (U/g)	Optimal pH and temperature	Antioxidant activity			
			DPPH (%)	ABTS (%)	Hydroxyl (%)	O_2_^−^^●^ (%)
Alcalase	20,000	pH 9.0; 55 °C	74.9 ± 1.6 ^b^	45.5 ± 1.5 ^a^	56.1 ± 1.8 ^a^	37.6 ± 1.2 ^a^
Trypsin	250,000	pH 7.5; 45 °C	66.4 ± 2.0 ^c^	35.0 ± 1.8 ^b^	46.0 ± 1.6 ^c^	32.7 ± 1.6 ^b^
Papain	10,000	pH 7.5; 50 °C	78.4 ± 0.9 ^a^	14.2 ± 1.1 ^d^	39.4 ± 1.9 ^d^	23.5 ± 1.1 ^c^
Neutrase	50,000	pH 7.5; 45 °C	57.9 ± 1.7 ^e^	30.0 ± 2.1 ^c^	52.1 ± 1.2 ^b^	21.1 ± 1.7 ^c^
Pepsin	10,000	pH 3.0; 37 °C	80.2 ± 1.8 ^a^	10.6 ± 1.3 ^e^	48.8 ± 1.4 ^c^	31.3 ± 1.4 ^b^

DPPH (1, 1-Diphenyl-2-picrylhydrazyl), ABTS (2, 2’-Azino-bis), hydroxyl and O_2_^−^^●^ (superoxide radicals) were used to detect the antioxidant activity of different enzymatic hydrolysates. The concentration of all hydrolysates was 1 mg/mL, except in the O_2_^−^^●^ scavenging test, in which the concentration was 2 mg/mL ^1^. The results are shown as the means ± standard deviations (SDs), and different letters indicate significantly different values (*n* = 3) at *p* < 0.05 according to Duncan’s multiple range test.

**Table 2 antioxidants-08-00433-t002:** The antioxidant activity of the ultrafiltration fractions (F1–F4).

Ultrafiltration	F1	F2	F3	F4	Positive
DPPH (%)	75.4 ± 2.1 ^a^	61.0 ± 1.9 ^b^	62.8 ± 2.5 ^b^	38.4 ± 1.0 ^c^	74.4 ± 2.1 ^a^ _(GSH)_
ABTS (%)	47.4 ± 1.0 ^c^	36.8 ± 1.2 ^e^	41.3 ± 1.7 ^d^	69.7 ± 1.0 ^a^	58.2 ± 1.8 ^b^ _(V__C__)_ **^2^**
Hydroxyl (%)	50.2 ± 1.7 ^c^	49.6 ± 1.5 ^c^	52.4 ± 1.8 ^b^	68.8 ± 1.2 ^a^	69.2 ± 1.6 ^a^_(GSH)_
O_2_^−^^●^ (%) **^1^**	32.2 ± 1.1 ^c^	37.8 ± 1.3 ^b^	37.3 ± 1.0 ^b^	30.8 ± 1.4 ^c^	48.3 ± 1.3 ^a^_(GSH)_

The concentration of all hydrolysates was 1 mg/mL, except in the O_2_^−^^●^ scavenging test, in which the concentration was 2 mg/mL ^1^. The concentration of ascorbic acid (Vc) was 1 mg/mL ^2^. The concentration of glutathione (GSH) was 0.1 mg/mL and that of Trolox and ascorbic acid was 0.01 mg/mL. The results are shown as the means ± standard deviations (SDs), and different letters indicate significantly different values (*n* = 3) at *p* < 0.05 according to Duncan’s multiple range test.

**Table 3 antioxidants-08-00433-t003:** Partial natural charge distribution of the five peptides was analyzed by natural bond orbital (NBO).

FSAP		PVETVR		QEPLLR		EAAY		VLRPPLS	
Atom	Charge	Atom	Charge	Atom	Charge	Atom	Charge	Atom	Charge
N1	−0.88759	N1	−0.69881	N1	−0.88921	N1	−0.91883	N1	−0.88832
H2	0.38462	H2	0.40074	H2	0.37610	C2	−0.14216	H2	0.37434
C16	0.69191	C7	0.71086	O8	−0.69341	C5	0.83494	N24	−0.68781
O17	−0.66103	O8	−0.67434	N9	−0.83729	O6	−0.59491	H25	0.42348
N18	−0.65131	N9	-0.6504	H10	0.44054	O7	−0.71113	N27	−0.80799
H19	0.41982	H10	0.42203	C19	0.83827	N10	−0.66883	H28	0.34732
C20	−0.1693	O31	−0.75823	O20	−0.62187	H11	0.44739	N29	−0.88529
O22	−0.75366	H32	0.47055	C22	0.71839	C14	0.69475	H30	0.39771
H23	0.48868	C33	−0.70866	O23	−0.68693	O15	−0.65216	C32	0.72417
C24	0.68167	C34	0.69293	C38	0.69139	C35	0.32133	O33	−0.67817
O25	−0.65711	O35	−0.66451	O39	−0.63917	O36	−0.68939	C39	0.72043
N26	−0.63958	N50	−0.66717	N55	−0.68442	H37	0.48477	O40	−0.67631
H27	0.44129	H51	0.43862	H56	0.41222	C38	0.82460	N41	−0.48518
C30	0.72173	N53	−0.8806	N58	−0.89905	O39	−0.60017	C46	0.70801
O31	−0.69570	H54	0.40155	H59	0.40002	H46	0.39801	O47	−0.67163
C37	0.81978	N56	−0.85604	N61	−0.79998	H47	0.38678	O61	−0.76549
O38	−0.58935	H57	0.34416	H62	0.35620	O59	−0.70009	H62	0.48539
H42	0.38653	C58	0.83508	O109	−0.70815	H60	0.51537	C63	0.84110
O57	−0.70917	O59	−0.59617	H110	0.49871	-	-	O64	−0.62418
H58	0.51168	O100	−0.71190	H111	0.49870	-	-	O118	−0.71865
-	-	H101	0.49801	-	-	-	-	H119	0.53081

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
