# Peer review of "In Vitro and In Silico Antioxidant Activity of Novel Peptides Prepared from Paeonia Ostii ‘Feng Dan’ Hydrolysate"

_antioxidants, 2019, doi:10.3390/antiox8100433_

Round 1
Reviewer 1 Report
Dear authors
After reading your manuscript, some questions have been raised:
in the document (lines 17, 64, 74) you use active site, chemical reaction sites and bioactive as synonymous. However, I think the most appropriate expression is bioactive because active site is the local in the enzyme where the substrate(s) binds and gives rise to product(s), reaction sites do not define the peptide fraction that have antioxidant activity, so bioactive peptide is more appropriated, in my opinion; in line 33 you start by defending oxidative stress, however it is known that it can cause damage to the body, but it can also do no damage at all. The sentence needs to be reworded; in line 49 use BDE without first mentioning throughout the text what the abbreviation means; In line 88, I do not understand which enzyme solution they refer to; In line 89, I suggest that instead of writing "... to terminate the hydrolysis reaction .." write "to terminate the enzymatic reaction"; Section 2.3 does not describe which markers are used nor the method for staining the SDS-PAGE; Section 2.4.2, was ion exchange chromatography isocratic as described? No change in pH or ionic strength? Section 2.4.3, the chromatography was only performed at 220 nm? Why was the detection at 280 nm not performed either? Section 2.5 is an analysis of the results of Section 2.4.4. No chemical peptide synthesis was performed here ... or was it? Line 204, GSH concentration? Section 2.8, you do not describe the conditions of cell culture, nor the number of L929 cells used in the assay, the data are very incomplete. Are the concentrations you present the final concentrations in each well? What were the negative and positive controls? In line 223, , I suggest that instead of writing "The DH pf PoSMP five...” write ”The DH pf PoSMP of five …"; Results and Discussion, on line 220, you should round the deviation from 0.27 to 0.3. The correlation between the degree of hydrolysis and the SDS-PAGE results is not clear. In the graph (Figure 1A) there is, as is said in the text, no decrease, but a stabilization. Lines 223 and 224 do not describe the hyperbolic behaviour of the results obtained; In figure 1B, AHs, THs, PaHs, NHs, PeHs ,it must be indicated that they are abbreviations of alcalase, trypsin, papin, neutrase, pepsin hydrolysates; Line 240, What is the justification for indicating alkalase as the best? Why not consider pepsin? Section 3.2, how were the peptide masses determined? Section 3.3.2, we cannot say that they are 7 components, but seven peaks, because there are compounds with similar characteristics that can by eluted at the same time and give rise to an overlapping peak! In section 3.3.3, in line 292, we cannot say that they are the major components, but rather the most intense peaks, in the length from which they are used for detection. In figures 2 and 3 attention to the repetition of the letters (a), (A), (b), (B)… Use uppercase or lowercase as in picture caption; Line 379, I suggest that instead of writing "...of hydrolysis of alcalase.…” write "...of hydrolysis by alcalase.…”; In abbreviation section put all abbreviations used in the document Best regards
Reviewer 2 Report
Dear Authors, you should address my recommendations highlighted across the text.

Round 2
Reviewer 1 Report
Dear Editor
In my opinion the article is ready for publication. Congratulations.
Best regards